# Emojis influence autobiographical memory retrieval from reading words: An fMRI-based study

Christos Chatzichristos[1,2☯]*, Manuel Morante[1,3☯], Nikolaos Andreadis[4], Eleftherios Kofidis[1,5], Yiannis Kopsinis[6], Sergios Theodoridis[1,3,7]

1 Computer Technology Institute & Press "Diophantus" (CTI), Patras, Greece, 2 STADIUS, Department of Electrical Engineering (ESAT), Leuven, Belgium, 3 Department of Informatics and Telecommunications, National and Kapodistrian University of Athens, Athens, Greece, 4 Bioiatriki SA, Athens, Greece, 5 Department of Statistics and Insurance Science, University of Piraeus, Greece, 6 Libra MLI Ltd, Edinburgh, United Kingdom, 7 Chinese University of Hong Kong, Shenzhen, China

☯ These authors contributed equally to this work.
* cchatzic@esat.kuleuven.be, chrichat@hotmail.com

**Data Availability Statement:** The data underlying the results presented in the study are available on Openneuro. https://openneuro.org/datasets/

## Abstract

Advances in computer and communications technology have deeply affected the way we communicate. Social media have emerged as a major means of human communication. However, a major limitation in such media is the lack of non-verbal stimuli, which sometimes hinders the understanding of the message, and in particular the associated emotional content. In an effort to compensate for this, people started to use emoticons, which are combinations of keyboard characters that resemble facial expressions, and more recently their evolution: emojis, namely, small colorful images that resemble faces, actions and daily life objects. This paper presents evidence of the effect of emojis on memory retrieval through a functional Magnetic Resonance Imaging (fMRI) study. A total number of fifteen healthy volunteers were recruited for the experiment, during which successive stimuli were presented, containing words with intense emotional content combined with emojis, either with congruent or incongruent emotional content. Volunteers were asked to recall a memory related to the stimulus. The study of the reaction times showed that emotional incongruity among word+emoji combinations led to longer reaction times in memory retrieval compared to congruent combinations. General Linear Model (GLM) and Blind Source Separation (BSS) methods have been tested in assessing the influence of the emojis on the process of memory retrieval. The analysis of the fMRI data showed that emotional incongruity among word +emoji combinations activated the Broca's area (BA44 and BA45) in both hemispheres, the Supplementary Motor Area (SMA) and the inferior prefrontal cortex (BA47), compared to congruent combinations. Furthermore, compared to pseudowords, word+emoji combinations activated the left Broca's area (BA44 and BA45), the amygdala, the right temporal pole (BA48) and several frontal regions including the SMA and the inferior prefrontal cortex.

ds002711 DATASET DOI 10.18112/openneuro.
ds002711.v1.1.0.

**Funding:** The research leading to these results was
funded by the European Union's H2020 Framework
Programme (H2020-MSCA-ITN-2014) under grant
agreement No.~642685 MacSeNet. The funders
had no role in study design, data collection and
analysis, decision to publish, or preparation of the
manuscript. The funders provided support in the
form of salaries for authors N.A and Y. K., did not
have any additional role in the study design, data
collection and analysis, decision to publish, or
preparation of the manuscript. The specific roles of
these authors are articulated in the 'author
contributions' section.

**Competing interests:** We would like to declare that
none of the authors and co-authors have any
competing interest. None of the authors or co-
authos have served or serves on the editorial board
of Plos-One, have acted as an expert witness in
relevant legal proceedings and have sat or currently
sits on a committee for an organization that may
benefit from publication of the paper. The authors
N.A. and Y.K. have been funded (in the form of
salaries) from the commercial affiliations of
Bioiatriki, SA and Libra, MLI Ltd, respectively. This
funding of the co-authors does not alter our
adherence to PLOS ONE policies on sharing data
and materials.

## Introduction

Communication plays an essential role in our everyday life and draws on both verbal (e.g., speech) and nonverbal (e.g., gestures, facial expressions, the tone of the voice) cues to convey information. In the last few decades, the use of computers, smart-phones and Internet has radically revolutionized communication. The Face-to-Face Communication (FFC) has been replaced to a large extent, by the so-called Computer-Mediated Communication (CMC) [1]. The number of emails sent per day (estimated at 267 billion [2] for 2017) and the fact that half of the world's population owe an email account are only some of the indications of the extent of this transformation. CMC is an effective tool for communicating verbal messages, yet it lacks the subtle nonverbal cues of FFC. In FFC, multi-modal signals are naturally produced (auditory and visual at least) while in CMC only visual-verbal cues are available, making typed messages on a computer screen appear emotionally neutral. The absence of non-verbal cues may reduce the communication efficiency and this may cause a misinterpretation of the nature of the message and the intention of the sender [3]. Hence, CMC needs to become more elaborate and less prescriptive and adapt to social needs such as to convey emotion.

The use of emoticons is a possible way to convey emotion in text communication and compensate for the lack of nonverbal communicative cues. According to its formal definition, the word *emoticon* is a blend of the words "emotion" and "icon". In general, emoticons are formed by a combination of keyboard characters to resemble facial expressions, e.g., :-) for happy face and :-( for sad face.

Emoticons function similarly to facial cues, tone of voice, and body language in FFC and augment a written message with non-verbal elements [1]. CMC is evolving rapidly, as Instant Messaging (IM) tends to get over emails as the primary CMC means [4]. Following the emergence of IM, emoticons have evolved to emojis, which enhance even more the expressiveness of messages. Emoji stands for pictograph, and the term originates from the Japanese words *e* "picture" and *moji* "character" [5]. Emojis are graphic symbols with shapes and colors, and they have become extremely popular in IM applications. Even companies have started creating their own emojis for marketing reasons [6], while an Emojipedia [7] has even been created and the World Emoji Day [8] has been established.

Although research of the use and interpretation of emoticons and emojis is still in its infancy, it has already been shown that they disambiguate the communicative intent behind messages [1], they can provide information concerning the user's personality [9] and they can even be used for the prediction of the stock market [10]. Merging words with emojis can also produce semantic congruity effects. It has been demonstrated that adding emojis in sentences may produce semantic integration. However, it can also result in ambiguity or misinterpretation if they are incongruent with the context of the text [1].

The vast majority of the existing studies on emoticons and emojis rely only on behavioral data. However, a combination of such data with neurophysiological findings should provide more reliable evidence for the relevance and the role of the emoticons/emojis, as a type of non-verbal communication within the CMC. The most commonly used imaging modalities for monitoring the brain activation and the neurophysiological responses are electroencephalography (EEG) and functional Magnetic Resonance Imaging (fMRI). Most of the studies with neurophysiological findings associated with emoticons or emojis are based on EEG, due to its easier accessibility and lower cost. However, EEG is known to suffer from poor spatial resolution, limited by the number of electrodes employed and the resistive properties of the extra-cerebral tissues. Furthermore, due to the fact that electrodes are more sensitive to neural activations that occur closer to the scalp, determining the exact location of activations that take place in deeper areas is more challenging [11]. On the other hand, fMRI provides a better

spatial description of the activation patterns within the brain and the potentially involved functional brain networks. However, the time resolution of the fMRI modality is much lower compared to EEG.

It should be emphasized, that the observation of unreal or artificial scenarios associated with an emotion activates neurons from the same network that is activated when sensing that emotion [12]. For example, viewing paintings of people getting hurt or of damaged body members results in the activation of areas similar to those that are activated by our own sensation of pain. In view of this, emoticons and emojis, despite their obvious artificial nature, are highly likely to also contribute to the comprehension of the semantic content of a message [13].

Several EEG studies have pointed out that emoticons and emojis [14–16] bear an emotional content, similarly to words. Furthermore, emoticons evoke the N170 event-related potential An event-related potential (ERP) is the time-locked electrophysiological activation of the brain which results directly from a specific stimulus [11]. (ERP) that reflects the neural processing of real faces. When potentials evoked by real and schematic faces (emoticons) are compared to those elicited by other visual stimuli [14], the former show increased negative potential 130–200 ms (N200) after the presentation of the corresponding stimulus at lateral occipital sites and especially on the right-hand side of the brain. The N200 ERP, which had been also recorded intracranially (on the ventral plane of the temporal lobe), is modulated primarily by the holistic, face integration mechanism located in the posterior lateral parts of the fusiform gyrus.

As mentioned before, unlike emojis, emoticons are made up of typographic symbols representing facial features and look like a sideways face. In an attempt to examine whether emoticons are processed configurally or featurally by the neurons, the neuronal response to inverted emoticons (with the punctuation marks rotated from the "natural face" position) has been studied in [17]. Emotions, natural faces and strings of typographic characters were presented to the volunteers either at their "natural" position or rotated (90 degrees). The N170 ERP was seen to be larger and/or delayed for inverted real faces than for upright ones. It was shown, in [17] that the inverted emoticons produce reduced N170s (contrary to real faces) indicating that the punctuation marks are understood with their default typographic meaning of colon, hyphen or parenthesis, e.g., if :-) is inverted to )-:, and thus no face is recognized, in contrast to natural faces, which were correctly recovered, independently of their relative orientation. That study showed that emoticons are perceived as faces only through configural processes. This is becaus, once they are inverted and the configuration is disrupted, the parts of the emoticon are no more perceived as facial features. This is a significant difference between emoticons and real faces, which may explain why no face-recognition areas have been observed in [18].

Furthermore, the study in [18] investigates the neuronal activation caused by the semantic content of emoticons, through three different comparative experiments: a) face images and non-face images, b) emoticons and non-emoticons, and c) a number of sentences without emoticons and some with emoticons at the conclusion of the sentences, where the emoticons were either congruent or incongruent with the sentence. Experiment c) was replicated in [19] with similar results. According to these studies, the right fusiform gyrus and the posterior cingulate gyrus were activated in response to photographs of faces in the first experiment but not with the appearance of emoticons. The right fusiform gyrus has been known to be activated during the perception of faces [20], while the posterior cingulate gyrus was reported in experiments with discrimination between "happy" and "sad" emotional nouns and adjectives [21]. Nevertheless, the facial representation of emoticons was not significant enough to activate these areas. Although there is no clear evidence that emoticons activating the right fusiform gyrus, it has been demonstrated, e.g., [18, 19], that seeing emoticons activates emotional valence detection even if they are not perceived as faces. The right inferior frontal gyrus, which

is associated with the emotional valence decision task, appears also to be activated with the use of emoticons, even when involving congruent and incongruent ones (experiment c) [18, 19]). The conclusions of the study in [18, 19], as it is stated by the authors, can be summarized as follows: i) emoticons are a kind of nonverbal information ii) brain sites dealing with both verbal and nonverbal information are activated more strongly when emoticons are added to sentences, than in the case of plain text and iii) since no significant activation was detected at the fusiform gyrus or the posterior cingulate gyrus, they assume that emoticons do not carry clear semantic content, in contrast to nouns or adjectives.

Furthermore, the neural responses of the brain to incongruent stimuli or conflict resolution have been studied for years. For example, several works have examined the response of the brain to semantic conflicts such as the Stroop word-color effect [22–24]. Put succinctly, the classical Stroop word-color effect relies on the observation that naming the ink color of a word takes longer and is more prone to errors if the ink does not match the color of the word [25] (e.g., naming the ink color of the word *blue* takes more time if the ink of the word is not blue). In those studies, the analysis of the fMRI data showed that, compared to congruent word-color stimuli, incongruent combinations caused activations of the paracingulate gyrus, the middle and inferior frontal gyrus (including BA47), the lateral occipital, the precuneus and the anterior cingulate. Among these areas, the anterior cingulate gyrus plays a crucial role in ambiguity resolution, as [26, 27] also reported. On the other hand, studies involving incongruent and congruent gestures [15, 28, 29], emoticons [16] and cartoon clips [30] have been conducted with the use of EEG and the investigation of the ERPs. Besides the well-known and studied sensory or motor ERPs, there is also a large group of language (cognitive) related ERP components. The most researched among them is N400, a negative-going deflection that peaks around 400 ms post-stimulus, mainly located over the centro-parietal region, which comprises a response to semantic expectancy violations. The studies detected a much larger N400 ERP in the cases where incongruent events (words + gestures/ emoticons/cartoon clips) were presented. Furthermore, as it was reported, the participants required significant more time to respond to incongruent events than to congruent ones, in both EEG and fMRI studies.

The existing related works involving biomedical imaging data [15, 19] and behavioral data [31], as explained before, use a combination of sentences and emoticons in order to test their influence in reading emoticon-embedded text. The initial assumption of our study is that since the emoticons influence the reading of complete sentences (due to their emotional content), they will also influence the memory retrieval process.

The Autobiographical Memory Test (AMT) [32, 33] is a cue-word paradigm and therefore it is ideally suited to fMRI studies. Emotion-laden and neutral cue words are shown to subjects, who are then asked to use each cue as an initial point to recall a related memory. A number of reviews have indicated that Autobiographical Memory (AM) retrieval is associated with activation in the hippocampus, parahippocampal gyrus, lateral temporal cortices, temporo-parietal junction, lateral and medial prefrontal cortex, thalamus, and cerebellum [32, 34–38].

We postulate that the presentation of emojis with a significant emotional content following the presentation of written words, which have also a significant emotional charge, will either facilitate (in case of congruent emotional charge) or interfere (when the word and emoji have incongruent charge) with the retrieval of the particular memory. We focused our study on combinations with words rather than sentences, to avoid the presence of a specific context, since each sentence involves a particular context that may narrow its potential emotional content [39, 40]. For example, the sentence "My aunt had a heart attack and we went to the hospital" plus a happy emoji is much more restrictive to the memory retrieval than the single word

"hospital" with a happy emoji (incongruent cases) This has been also confirmed by the volunteers that selected the words. For example, the word hospital used in the sentence, mentioned above, for all the volunteers brought up a negative memory. The word hospital alone for some of the volunteers brought up happy memories—e.g. the birth of a child.

In this paper, we investigate the cognitive and behavioral effect of combination of words and emojis on AM retrieval (and word processing), in an effort to determine how emojis complement the written text and their effect on the emotional content of the message. Furthermore, another goal is to test whether the findings of [18, 19] for emoticons are valid for emojis too. To the best of our knowledge, this is the first time that congruent and incongruent combinations of words and emojis have been studied using fMRI.

## Materials and methods

An adapted version of the AMT, has been used to test for the influence of emojis on the neural modulation (activated brain areas) of autobiographical memory retrieval.

### Selection of words

In order to select appropriate Greek words to be used as cue words, we followed the next four-step procedure:

- *Step 1*—A group of 20 volunteers (14 females, mean age = 23.6±3.2 years) were asked to report 10 Greek nouns with emotional content and 10 without. Eventually, a total number of 315 different nouns with and without emotional content (excluding repetitions of course) were collected.

- *Step 2*—A new group of 24 volunteers (18 females, mean age = 28.3±6.3 years) were instructed to rate these 315 words according to their personal emotional content from 1 to 10, using a Likert scale (1 for the most significantly emotional content and 10 for the least one), through the SurveyMonkey's software [41]. According to this evaluation, we selected the 49 words that had a mean score of 3 or less.

- *Step 3*—A new group of 43 volunteers (25 females, mean age = 22.4 ±5.8 years) were instructed to rate the selected 49 nouns for the valance, arousal, imageability, and familiarity, again using SurveyMonkey's software (using the same Likert scale from 1 to 10). As stated in [42], arousal and valance are two of the three primary independent dimensions of emotions. Those three dimensions are: valence or pleasure (positiveness-negativeness / pleasure-displeasure), arousal (active-passive) and dominance (dominant-submissive). The latter is of no interest in this study.

Eventually, using these results, two groups of 6 words were created, namely, those that exhibit the biggest difference in valance (most positive and most negative words) and simultaneously have similar mean values of arousal, imageability, and familiarity, to avoid secondary effects. Table 1 shows the selected words. Valence ratings were significantly higher for positive (mean ($\mu$) = 7.67, standard deviation ($\sigma$) = 0.35), compared to negative words ($\mu$ = 2.20, $\sigma$ = 0.15; $t$ = −35.16, $p < 0.001$), hence the two groups were discriminative. Positive and negative words did not differ in arousal ($\mu$ = 3.55, $\sigma$ = 1.00; $\mu$ = 4.44, $\sigma$ = 1.16, for positive and negative words respectively, with $t$ = 1.43, $p$ = 0.18), imageability ($\mu$ = 5.84, $\sigma$ = 1.63; $\mu$ = 6.78, $\sigma$ = 1.13, for positive and negative words respectively, with $t$ = −1.16, $p$ = 0.27) and familiarity ($\mu$ = 5.76, $\sigma$ = 0.83; $\mu$ = 6.29, $\sigma$ = 0.77, for positive and negative words respectively, with $t$ = 1.15, $p$ = 0.28) ratings.

**Table 1. Words selected.**

|  | Word | Translation | Val. | Famil. | Imag. | Ar. |
|---|---|---|---|---|---|---|
| Positive | Έρωτας | Love | 1.49 | 7.62 | 7.66 | 4.39 |
|  | Συντροφιά | Companionship | 2.09 | 6.07 | 7.61 | 5.43 |
|  | Ελπίδα | Hope | 2.18 | 5.50 | 5.00 | 4.49 |
|  | Ζωή | Life | 2.19 | 7.29 | 5.89 | 3.91 |
|  | Ελευθερία | Freedom | 2.33 | 5.37 | 4.89 | 4.88 |
|  | Αλήθεια | Truth | 2.39 | 6.56 | 3.76 | 5.53 |
| Negative | Νοσοκομείο | Hospital | 7.26 | 5.91 | 8.37 | 3.11 |
|  | Μελαγχολία | Melancholia | 7.29 | 5.71 | 6.50 | 5.40 |
|  | Πόνος | Pain | 7.69 | 6.64 | 6.89 | 2.97 |
|  | Αρρώστια | Sickness | 7.71 | 6.53 | 7.21 | 3.11 |
|  | Απελπισία | Hopelessness | 7.91 | 5.35 | 4.88 | 3.50 |
|  | Εγκατάλειψη | Abandonment | 8.29 | 4.56 | 5.41 | 3.22 |

## Selection of emojis

In the study described in [5], for one of the first and most well-known emoji sentiment lexicons, created from the context of 1.6 million tweets, the emotional content of the emojis was labeled based on their sentiment polarity; *positive*, *neutral* or *negative*, according to the emotional context of the tweets where they appeared. From this ranking, three positive and three negative face emojis were selected, with approximately the same mean (0.605 for positive and -0.594 for negative). Our aim was to confirm whether the rankings were valid also for Greek native speakers, since the Greek language was not included among the 13 languages used in the study. Therefore, 50 volunteers from the University of Athens, Greece, who had reported a frequent use of IM, were asked to rate these emojis according to their emotional content using a Likert scale (1 for the happiest content and 10 for the saddest content).

Table 2 shows the sentiment ranking [5] of each emoji selected and the mean score given by the 50 Greek students. Note that the "Smiling face with open mouth and closed eyes" and the "Smiling face with open mouth" received the same mean score, however we chose the latter since the standard deviation (stdev) of the votes for it was lower and it also had a higher

**Table 2. Sentiment ranking of the different emojis used in this study.**

| Emoji | Description | SR [5] | LS |
|---|---|---|---|
| 😃 | Smiling face with open mouth | 0.629 | 1.7 ± 0.6 |
| 😆 | Smiling face with open mouth and closed eyes | 0.564 | 1.7 ± 1.1 |
| 😌 | Relieved face | 0.623 | 3.5 ± 2.4 |
| 😕 | Confused face | −0.601 | 5.9 ± 3.2 |
| 😒 | Unamused face | −0.591 | 6.7 ± 1.2 |
| 😩 | Weary face | −0.591 | 8.8 ± 0.9 |

SR: Sentiment Rank; LS: Likert Score

emotion. For the negative face emojis, the "Weary face" was selected as it had been ranked as the saddest one. We opted to select only one positive and one negative emoji, because images with similar emotional content activate similar brain functional networks [43].

The first two columns show the used emojis and their description. The third column includes the emotional ranking [5] while the last column provides the Likert score given by Greek students.

### Participants

Twenty volunteers (8 females) participated in the study (age $\mu$ = 37.8 years, $\sigma$ = 11.0, range 24–67). All participants were free of medication. Among the exclusion criteria, we included: present or previous psychiatric disorder, high levels of myopia (in case they could not wear eye lenses) and native language other than Greek. Exclusion criteria specific to the fMRI scanning were: spectacles, heart pacemaker, mechanical heart valve or any mechanical implants, potential pregnancy, and claustrophobia. All participants gave written informed consent prior to the exam (Supplementary material). The study was undertaken with ethics approval granted by the University of Surrey, U.K. (the coordinating partner of the Marie Skłodowska-Curie Innovative Training Network (MacSeNet) funding this research) and its Psychiatric Research Ethics Committee. During the exam, three participants quit (2 females) and the scan of the first volunteer (male) was considered as a pilot scanning. Furthermore, after the study of the behavioral data, we noticed that one of the participants (male) presented a considerable number of missed stimuli. Therefore, we decided to exclude also this participant from the analysis, since it seemed that he was not properly engaged in the task. Thus, the final group consisted of 15 participants (6 females, age $\mu$ = 34.5 years, $\sigma$ = 8.0, range 24–57).

### fMRI task design

The experimental task used in this study consists of a pseudo block design The experimental task design and presentation was performed using the software Nordic Aktiva [44]. which contains a total number of 24 blocks, each block consisting of the presentation of an emotional stimulus cue (for 15 seconds), a fixation cross (for 5 seconds), a pseudoword (for 10 seconds) and a second cross (for 8-10 seconds) and lasts in total 38–40 seconds. Fig 1 shows an example of such as block. In total, the experimental run lasted approximately 18 minutes per subject. The cues were presented visually on screen in a fixed order. The target stimuli were composed of randomly alternating positive and negative words combined with happy or sad emojis. As documented in [6], emojis that ensemble faces or gesture do not replace any syntactic element of the sentence, hence, this is the reason that they are placed at the end of the message, acting as adverbs and expressing mood. Each word was presented twice, once with the happy emoji and once with the sad one. Participants were asked to recall a specific memory, in response to

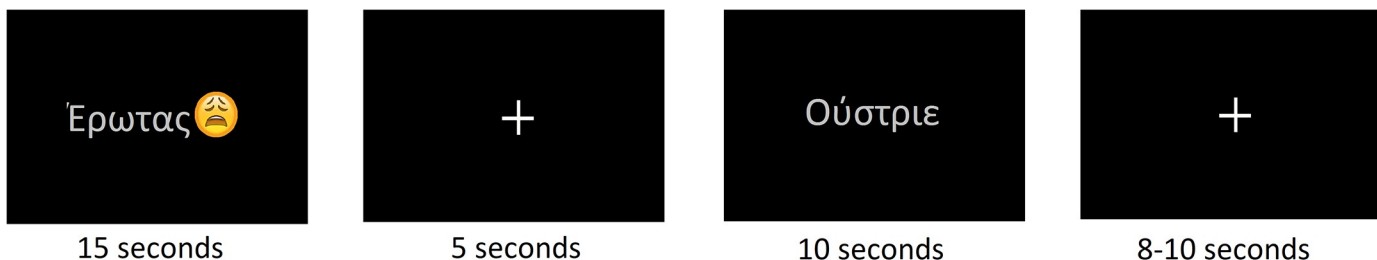

**Fig 1. A block of the experimental task.** An emotional stimulus (Έρωτας—Love and sad face) followed by a cross, a pseudoword (Ούστριε) and a second cross (with random timing).

each cue combination of a word and an emoji. They were asked to press a button as soon as they had retrieved a specific memory. Response time, i.e., the amount of time that elapses from the presentation of the cue until the participant presses the button was recorded, as an indicator for the memory-retrieval speed, in order to be used in the analysis of the behavioral data. After each word, a pseudoword (e.g., Ρτο, Αρετουντρες, Τρεφι) appeared on the screen and participants were asked to press a button as soon as they had silently counted the number of letters in the pseudoword. This was done in an attempt to disengage the participants from the memory retrieved from the previous stimulus. The length (number of letters) of the pseudowords matches that of the actual words and the order of their appearance was random.

Neuronal activation was modeled using four different states, namely: a) happy word followed by happy emoji (denoted HH), b) happy word followed by sad emoji (HS), c) sad word followed by sad emoji (SS), and d) sad word followed by happy emoji (SH). The states HH and SS are the congruent states (same emotion for word and emoji) and the states HS and SH are the incongruent ones. We have respected the following rules:

- At most three consecutive words or emojis of the same emotion (happy or sad) can appear in consecutive stimuli.

- At most two successive stimuli of the same state (e.g., HH, HH) are allowed and at most three in every eight stimuli.

- Every eight stimuli, all the four different states appear at least once.

- Every eight stimuli, at most five stimuli of the same group appear (congruent or incongruent).

Those rules were set in an attempt to retain the same time-courses among subjects but also include in the experiment certain degree of randomness similarly to event-related fMRI designs (the succession of the events was random, and the duration of the block was random). Two different stimuli sequences were randomly generated, according to the set of rules specified above. Each sequence was used on half of the volunteers. Following the creation of the first sequence, an extra rule was set for the second one: the same state condition should appear in every position of the sequences (e.g., each HH stimulus was replaced by an other HH, appearing with the same timing). This rule was set to allow, besides the general linear model (GLM) [45, 46], the use of certain blind tensorial methods [47, 48]. The application of data driven methods in multi-subject cases assumes the same (or at least very similar) time-course per subject, meaning that stimuli of the same type are presented to every subject at the same time The same time-course per subject is "translated" to a multi-linearity constraint in tensor decompositions. The multi-linearity constrained can be alleviated in more flexible models like PARAFAC2 and BTD2 [49].

Prior to the scanner experiment, a training session took place, where participants practiced with both a congruent and an incongruent combination of one sad and one happy word and were encouraged to keep searching for a memory until they reported a specific one. All participants were debriefed after exiting the scanner and were asked to report their experience. To maintain consistency among experiment realizations and to guarantee that all the participants were properly engaged in the realization of the experiment, we discarded the experiment of any participant who failed to respond to 4 or more stimuli (omissions).

## fMRI data collection

The scanning was performed in the facilities of BioMed (Vioiatriki S.A.) in Athens, Greece, with the aid of a whole body 3 Tesla GE MR750 scanner, with a standard quadrature birdcage

head coil. The structural scans were acquired with a 3-dimensional (3D) T1-weighted BRAVO sequence for each subject (repetition time [TR] 6.8 ms, echo time [TE] 2.6 ms, flip angle 10˚, 1.2 mm$^3$ isotropic voxels, image size 512 × 512 × 280; elliptical sampling, coronal orientation) and T2-weighted FLAIR sequence (repetition time [TR] 10 s, echo time [TE] 118.608 ms, flip angle 160˚, 3 mm$^3$ isotropic voxels, image size 512 × 512 × 36; elliptical sampling, coronal orientation) to facilitate later co-registration of the fMRI data into standard space. Functional images were acquired in the form of T2*-weighted echoplanar imaging slices (TR 2 s, TE 45 ms, flip angle 10˚, 4 mm$^3$ isotropic voxels, image size 64 × 64 × 33; elliptical sampling, coronal orientation). A total of 540 volumes per subject were acquired during the AMT. Before the collection of the data of the pseudo-block design described in the previous section, 225 volumes of T2*-weighted resting state data were collected (TR 2 s, TE 45 ms, image size 64 × 64 × 33, 4 mm$^3$ isotropic voxels) for future research.

## Behavioral data analysis

As mentioned previously, the participants were asked to press a button once they had managed to retrieve a clear memory; the time between the presentation of the stimulus and the pressing of the button was recorded. In case the subject did not press the button in the allocated time, the stimulus was considered as missed.

After examination of the data acquired from the first six subjects, it was noted that one of the stimuli (incongruent) had not been presented due to the malfunctioning of the workflow software. Therefore, we opted to remove the same stimuli (at the corresponding time point) for all the subjects for both sequences. Furthermore, before the statistical analysis of this dataset, we manually excluded all the data of the first stimuli (congruent) for all of the participants, to guarantee that they were properly engaged into the task. In total, the response times of 330 stimuli (165 congruent) of 15 subjects (22 stimuli per subject) were analyzed.

## fMRI data analysis

In this study, fMRI data preprocessing was carried out using FEAT FMRI Expert Analysis Tool (FEAT) version 6.00 from the FSL software package [46]. Motion correction was performed using Motion Correction FMRI'S Linear Image Registration Tool (MCFLIRT) [50], followed by a slice-timing correction using Fourier space time-series phase-shifting. Then, brain volumes were extracted using Bet Extraction Tool (BET) [51] and brain images were smoothed with a 8 mm Full Width at Half Maximum (FWHM) Gaussian Kernel (twice the voxel size). Eventually, grand-mean voxelwise intensity was normalized over the entire 4D dataset by a single multiplicative factor followed by a highpass temporal filtering (Gaussian-weighted least-squares straight line fitting, with standard deviation $\sigma = 50.0$ s). Registration of the fMRI data to high resolution T1 and T2 images and to standard (MNI atlas) images was carried out using the FMRIB's Linear Imaging Registration Tool [50]. Time-series statistical analysis was carried out using FMRIB's Improved Linear Model (FILM) with local autocorrelation correction [52].

In the first-level analysis, each stimulus was modeled as a block according to its emotional charge (HH, SS, HS, SH and pseudowords), with duration equal to its presentation time. The specific task-related time courses were estimated using the canonical heamodynamic response function (HRF) [45, 53]. Temporal and dispersion derivatives were also included in the GLM model as extra regressors. Then, for each participant, we defined three different contrasts: a) word+emoji > pseudoword, b) incongruent (HS+SH) > congruent (HH+SS) and c) congruent (HH+SS) > incongruent (HS+SH). The first contrast aims to study the main functional brain areas involved in the processing of word+emoji stimulus during the memory retrieval

processing, independently of their emotional valence. We also introduced the contrasts b) and c) to investigate the implication of incongruent emotional combination of words and emojis.

## Results

### Behavioral data

From the study of the reaction time of the participants, a total number of 9 omissions ($\sim$ 2.5%) occurred. The vast majority of the missed stimuli (8 out of 9) were incongruent combinations (6 SH and 2 HS); only one was a congruent stimulus (SS). When interviewed after the scanning, those subjects who did not press the button on time reported that they did not manage to retrieve a clear memory for those cases, while in the cases that they did press the button they have.

In order to detect whether the emotional content of the emojis influenced the response time of the participants, we analyzed potential variations among congruent and incongruent stimuli, arising either from the emotional content of the words or from the emojis. Thus, the events were categorized into four different sub-groups according to their emotional charge, as mentioned before (HH, SS, HS and SH). Fig 2 displays the mean value of the time response of all the participants for each group and the error lines represent the standard deviation. An ANOVA test (see Table 3) revealed the existence of a significant difference at the $p < 0.05$ level for the four stimuli ($F(3, 339) = 3.902$, $p = 0.009$). On top of that, a Tukey test for mean separation (see Table 4) pointed out that there exists a significant difference between the congruent positive stimulus (HH) and incongruent stimulus, i.e., the difference in the response time between congruent and incongruent is mainly driven from the stimuli of the HH state. However, although the observed difference between SS and incongruent events in Fig 2 is small and not statistically significant, according to the obtained results from the Tuckey test, this

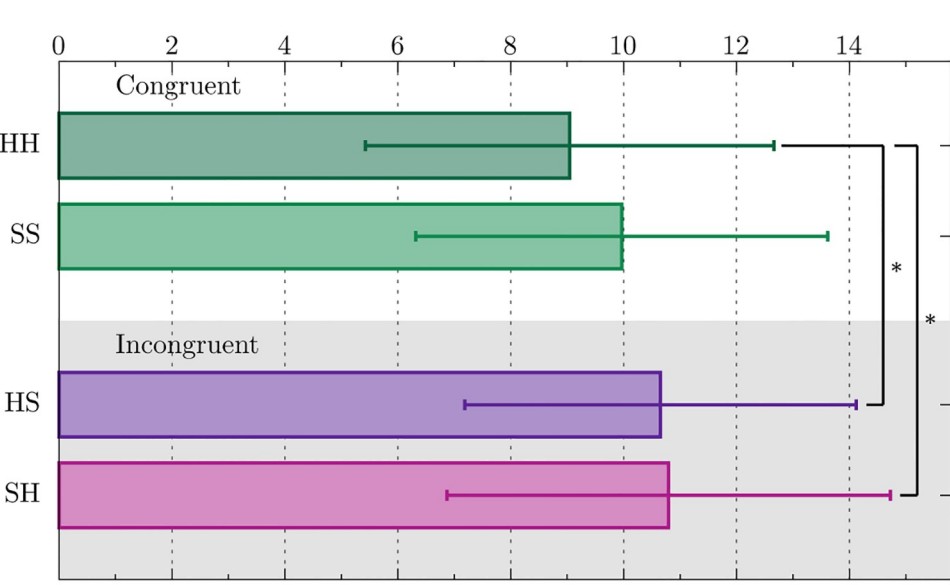

**Fig 2. Distribution of the timing of the responses of all subjects for the four different possible word and emoji combinations: HH, SS, HS and SH.** The bars represent the mean value, $\mu$, and the error lines the standard deviation, $\pm\sigma$, of each distribution of results. *Significant difference with $p \leq 0.05$ from the Tukey test.

**Table 3. Results of the ANOVA test for all the studied participants.**

| Test | SS | df | MSE | F-test | Sig |
|---|---|---|---|---|---|
| Between groups | 158 | 3 | 53 | 3.90201 | 0.009 |
| Within groups | 4582 | 339 | 14 | | |
| Total | 4740 | 342 | | | |

SS: Sums of Squares, df: degree of freedom

MSE: Mean Squared Error, Sig: Significance (*p* value)

**Table 4. Results from the Tukey mean separation test of the different types of stimuli.**

| Groups | | MD (s) | SE | *p* |
|---|---|---|---|---|
| HH | SS | -0.924 | 0.411 | 0.39 |
| | HS | -1.609 | 0.416 | 0.03* |
| | SH | -1.753 | 0.411 | 0.02* |
| SS | HH | 0.924 | 0.411 | 0.39 |
| | HS | -0.685 | 0.416 | 0.66 |
| | SH | -0.829 | 0.388 | 0.43 |
| HS | HH | 1.609 | 0.416 | 0.03* |
| | SS | 0.685 | 0.416 | 0.66 |
| | SH | -0.144 | 0.416 | 0.99 |
| SH | HH | 1.753 | 0.411 | 0.02* |
| | SS | 0.829 | 0.388 | 0.43 |
| | HS | 0.144 | 0.416 | 0.99 |

MD: Mean Difference, SE: Standard Error

* Significant ($p \leq 0.05$)

result may have been driven by the reduced sample size and the large variance of the participants' responses.

## fMRI data analysis with FSL

The statistical analysis of the FSL's results was carried out using FEAT. The obtained significant activation maps were thresholded for significance at $z > 3.09$ ($p < 0.001$), and they were corrected for Family-Wise Error at $p < 0.05$.

Table 5 lists all the significant clusters for the two studied contrasts (a) word+emoji > pseudoword and (b) incongruent > congruent. This table also shows the coordinates of the maximum peak activation, including that up to 4 subclusters, its anatomical correspondence within Brodmann Atlas [54], and its anatomical label. Fig 3 shows the significant activation maps for $z > 3.09$ ($p < 0.001$, FWE corrected) of the two studied contrasts.

Finally, regarding the study of the third contrast (c) congruent > incongruent, no significant activation patterns were detected, except for a small residual within the left ventricles, as it can be observed in the Supplementary material. Similarly, the single-subject analysis of this contrast did not provide any significant activation for any of the subjects.

## fMRI data analysis with BTD

In addition to the standard group analysis using FSL, we have also applied Block Term Decomposition (BTD) [55–57], a blind source separation method based on tensor decomposition.

**Table 5. Significant activation clusters from the contrasts of FSL.**

| Cluster Size | z-score | x | y | z | BA | Anatomical Labels |
|---|---|---|---|---|---|---|
| *word+emojis > pseudoword* | | | | | | |
| 14737 | 5.59 | -44 | 16 | -18 | 44,45,47,8 | L IPC, Broca's area, S frontal, SMA |
| | 5.26 | -46 | 26 | -6 | | |
| | 5.25 | -44 | 30 | -12 | | |
| | 5.21 | -4 | 20 | 58 | | |
| 13593 | 6.10 | 44 | -78 | -30 | 18,19,37 | R/L cerebellum, R/L secondary vis, I Occipital, L/R posterior fus |
| | 5.64 | 28 | -84 | -28 | | |
| | 5.57 | 22 | -76 | -30 | | |
| | 5.45 | -4 | -92 | -10 | | |
| 2273 | 5.49 | 42 | 28 | -14 | 38,45 | R M frontal, R temporal pole, pars triangularis |
| | 5.41 | 54 | 28 | 2 | | |
| | 5.13 | 54 | 26 | -2 | | |
| | 4.95 | 52 | 14 | -6 | | |
| 604 | 5.12 | -42 | -54 | 20 | 39 | L I parietal, angular |
| | 3.55 | -54 | -72 | 20 | | |
| 498 | 4.46 | -28 | -30 | -6 | 20,37,35 | L I fusiform, L hipocampal |
| | 3.96 | -14 | -20 | -18 | | |
| | 3.64 | -32 | -14 | -16 | | |
| | 3.35 | -32 | -22 | -14 | | |
| *incongruent > congruent* | | | | | | |
| 5716 | 4.68 | 52 | 30 | 16 | 45,44,47 | R IPC, pars triangularis/opercularis |
| | 4.63 | 46 | 44 | -14 | | |
| | 4.60 | 48 | 14 | 32 | | |
| | 4.24 | 46 | 20 | 18 | | |
| 4534 | 4.61 | -54 | 18 | 4 | 45,6,47,38 | L IPC, Broca's Area, SMA, L temporal pole |
| | 4.24 | -38 | 0 | 44 | | |
| | 4.17 | -38 | 2 | 50 | | |
| | 4.12 | -46 | 26 | -10 | | |
| 3099 | 4.57 | -4 | 26 | 40 | 32,8 | S frontal, anterior cingulate, SMA |
| | 4.30 | 6 | 38 | 44 | | |
| | 4.07 | -10 | 12 | 50 | | |
| | 4.06 | -2 | 12 | 48 | | |
| 2285 | 4.60 | 32 | -68 | 48 | 7,39,40,22 | R S/I parietal, angular, R S Temoral |
| | 4.30 | 42 | -64 | 40 | | |
| | 3.90 | 46 | -50 | 44 | | |
| | 3.79 | 48 | -56 | 22 | | |
| 2046 | 4.20 | -32 | -64 | 36 | 7,39 | L S/I parietal, angular |
| | 4.15 | -48 | -52 | 36 | | |
| | 4.06 | -28 | -72 | 42 | | |
| | 4.04 | -28 | -76 | 42 | | |
| 1613 | 3.92 | 26 | -72 | -12 | 18,19,37 | R secondary vis, R I Occipital, R pesterior fus |
| | 3.92 | 38 | -62 | -22 | | |
| | 3.86 | 28 | -76 | -18 | | |
| | 3.85 | 30 | -66 | -14 | | |

(*Continued*)

**Table 5.** (Continued)

| Cluster Size | z-score | x | y | z | BA | Anatomical Labels |
|---|---|---|---|---|---|---|
| 1175 | 4.08 | -10 | -80 | -30 | 18,19 | L cerebellum, L I Occiptial, L secondary vis |
| | 3.71 | -28 | -66 | -16 | | |
| | 3.55 | -44 | -70 | -10 | | |
| | 3.52 | -48 | -74 | -12 | | |
| 553 | 3.65 | -52 | -38 | -8 | 21 | L M Temporal |
| | 3.41 | -64 | -46 | 4 | | |
| | 3.36 | -66 | -46 | -4 | | |
| | 3.35 | -62 | -42 | -2 | | |

BA = Brodmann's Area, L = Left, R = Right, I = Inferior, M = Middle, S = Superior, IPC = Inferior Prefrontal Cortex, SMA = Supplementary motor area, vis = Visual Cortex, fus = fusiform gyrus

One of the main advantages of the fully blind methods is the fact that an explicit definition of the experimental task-related time-course is not required in order to obtain significant results; these methods learn both the parametric maps and the corresponding activation patterns (regressors of interest) from the data themselves. The BTD method relies on the assumption of low rankness in the spatial domain of the fMRI data in order to separate the different sources (each source is a combination of a spatial map and a time course) For completeness, Group Independent Component Analysis (GICA) [58] with the use of Infomax [59], which assumes independence instead of low-rankness of the sources, has been also tested. The results did not provide any extra physiological findings, hence they have been moved to the Supplementary material.

In order to identify the maps associated with our tasks of interest, we determined the specific sources of interest by selecting those whose time-courses present the highest correlation with the studied experimental tasks. In this case, we selected as a task of interest the experimental condition corresponding to all the word+emoji combinations (both congruent and incongruent) and the time course corresponding to the pseudowords, to further investigate the areas associated with this task. After the selection of the sources of interest, the obtained parameter maps were thresholded for statistical significance at $z > 3.09$ ($p < 0.001$). Table 6 lists all the significant clusters from sources that are most correlated with the (a) word+emoji and (b) pseudowords tasks. This table also includes relevant information regarding the obtained clusters as in Table 5. Finally, Fig 4 shows the significant activated maps associated with both tasks of interest from the BTD analysis.

## Discussion

The goal of this study was to determine how emotionally charged emojis affect the AM retrieval process in healthy humans. The aim was twofold: first, to investigate the main brain areas that participate in the word+emoji recognition during memory retrieval and, second, to determine the related effect of incongruent combinations of word+emoji in contrast to congruent combinations.

### Study of activated areas

Regarding the different brain areas activated during AMT, the study of the contrast word +emoji > pseudoword led to some notable results. First of all, the maximum significant activation of this contrast appears within the Broca's area (left BA44 and BA45), which agrees with

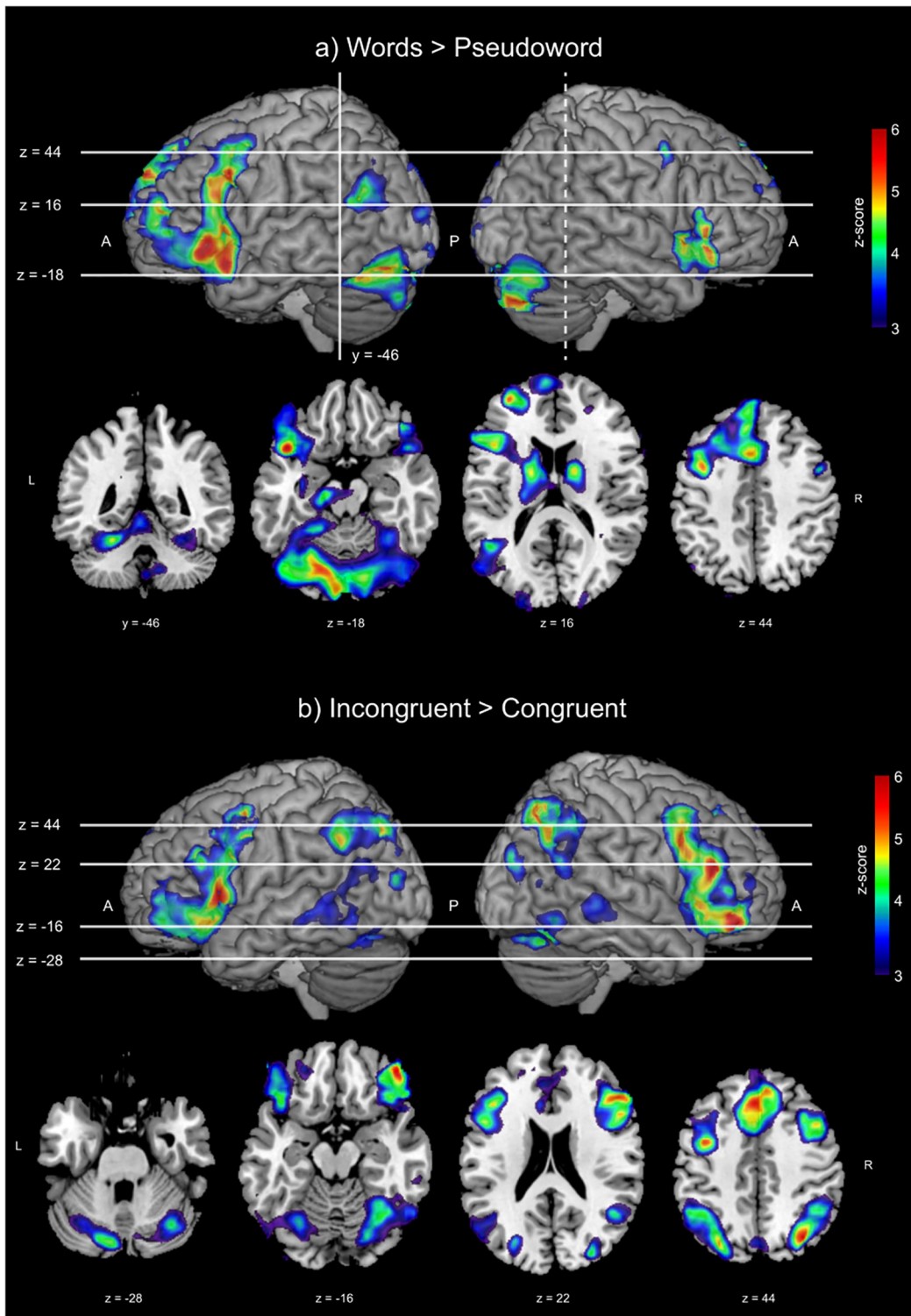

**Fig 3. Significant activation clusters from the group analysis for the two considered contrasts.** (a) Word+emoji > pseudoword (b) incongruent > congruent.

**Table 6. Significant activation clusters from the contrasts of BTD.**

| Cluster Size | z-score | x | y | z | BA | Anatomical Labels |
|---|---|---|---|---|---|---|
| *Word+eEmoji* | | | | | | |
| 3444 | 5.5 | 0 | 40 | 56 | 8,32,6 | S frontal, anterior cingulate, SMA |
| | 5.3 | 0 | 18 | 50 | | |
| | 5.2 | 0 | 16 | 66 | | |
| | 5.1 | 4 | 10 | 68 | | |
| 2934 | 4.9 | 12 | -92 | -2 | 17,18 | vis cortex, I occipital |
| | 4.8 | 12 | -94 | -6 | | |
| | 4.3 | -4 | -84 | -20 | | |
| | 4.1 | 16 | -84 | -26 | | |
| 1637 | 4.7 | -44 | 20 | 26 | 44,47,38 | L IPC, Brocas' Area, temporal pole |
| | 4.6 | -44 | 20 | 22 | | |
| | 4.3 | -44 | 20 | -14 | | |
| | 4.0 | -46 | 34 | -10 | | |
| 222 | 3.5 | 0 | -54 | 12 | 30 | posterior cingulate |
| | 3.3 | 0 | -44 | -6 | | |
| 19 | 3.1 | -44 | -72 | 30 | 39 | L I parietal |
| | 3.0 | -46 | -68 | 24 | | |
| *pseudoword* | | | | | | |
| 3399 | 6.0 | 34 | -62 | 56 | 7,40 | R S parietal |
| | 5.8 | 34 | -66 | 52 | | |
| | 5.1 | 10 | -78 | 52 | | |
| | 5.1 | 50 | -42 | 52 | | |
| 1394 | 5.0 | -34 | -54 | 56 | 7,40 | L S parietal |
| | 4.9 | -26 | -62 | 60 | | |
| | 4.8 | -22 | -66 | 60 | | |
| | 4.6 | -30 | -58 | 56 | | |
| 492 | 4.4 | 14 | -94 | 0 | 18 | R vis |
| | 3.2 | 14 | -94 | -16 | | |
| 319 | 4.1 | -18 | -90 | -20 | 18 | L vis |
| | 3.6 | -22 | -86 | -24 | | |
| | 3.6 | -14 | -94 | -16 | | |
| | 3.4 | -10 | -98 | -12 | | |
| 69 | 4.0 | 26 | 58 | 24 | 46,10 | R prefrontal |
| | 3.2 | 26 | 62 | 16 | | |
| 2 | 3.3 | 26 | -90 | -20 | 18 | R vis |
| 1 | 3.0 | -46 | -70 | -12 | 19 | L I occiptial |
| 1 | 3.0 | -42 | -38 | 40 | 40 | L S parietal |

Abbreviations as in Table 5.

the expected activation areas corresponding to natural brain network for language processing [60, 61]. Furthermore this contrast shows significant activation within the amygdala and several frontal regions, including BA8 and BA47 in both hemispheres. At this point, we would like to highlight the role of these areas, in particular, the amygdala and BA47, which has been know to participate in AMT [62, 63] and single-word reading [22–24, 64]. Finally, we have found significant activation within the right temporal pole, which plays a special role regarding emotion cognition and attention [65, 66]. Furthermore, Blair et al. in [67] showed that the

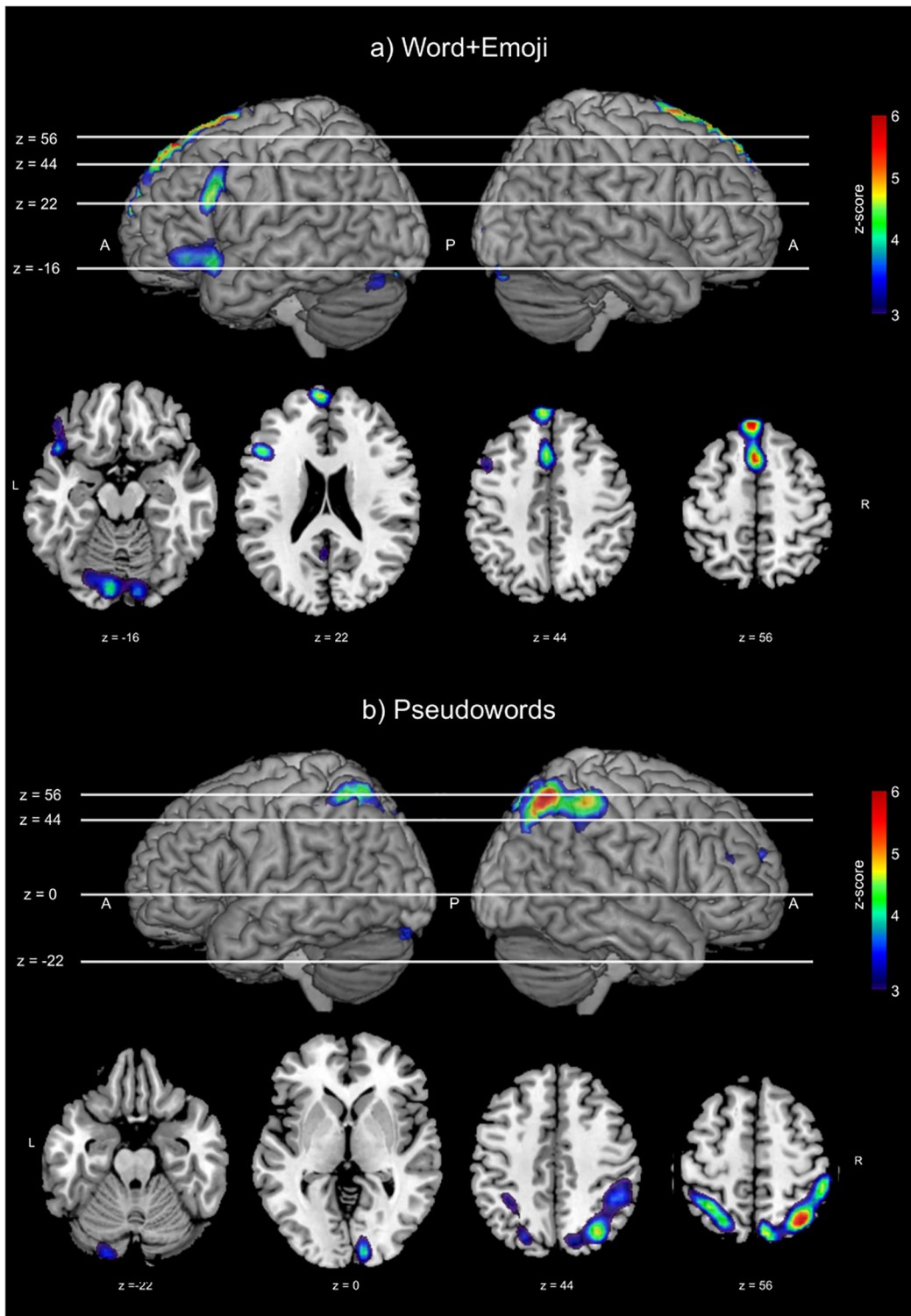

**Fig 4. Significant activation clusters from the analysis using BTD.** Threshold used for significant activation at $z > 3.09$ ($p < 0.001$).

right temporal pole in cooperation with the prefrontal cortex participates in the processing of distinct categories of negative facial expressions.

Therefore, the findings from this contrast provide evidence for the processing of word reading, emotion recognition, and memory retrieval. Moreover, our findings indicate that the presence of emojis within a reading-memory retrieval process trigger a more complex brain activation pattern than using only words [33, 37, 68]. These results contrast with previous studies on emoticons [18, 19], where no significant activation patterns regarding face-emotion recognition, e.g., fusiform gyrus, were observed.

### Effects of incongruent vs. congruent combinations

The analysis of the reaction times showed that emojis affect the reading-memory retrieval process. In particular, we have confirmed that the participants required significantly more time to retrieve a memory from an incongruent stimulus (i.e., word and emoji having contradicting emotional charge) than from a congruent positive stimulus (HH); see Fig 2. Moreover, the vast majority of the omitted responses (8 out of 9) correspond to an incongruent stimulus, due to the inability of the participant to retrieve a clear memory, as the subjects verbally reported after the completion of the experiment. These results agree with similar studies in [22, 23], where the participants required significantly more time to process incongruent stimuli compared to congruent.

Regarding the functional brain networks related to the processing of incongruent combinations of word+emoji, among the different activation patterns, the spatial map of the contrast incongruent > congruent exhibits significant activation within the Broca's area (BA44 and posterior BA45) in both hemispheres. Although Broca's area is widely recognized as an important brain area for language and speech processing, it also plays a major role in resolving a semantic conflict among representations [69, 70]. Furthermore, the study of this contrast also revealed a strong activation within the right/left IPC and the prefrontal Supplementary Motor Area (SMA). Interestingly, both IPC and SMA are regions that are all sensitive to stimulus-based language conflict [71]. Moreover, the presence of the BA47 in this contrast is also interesting. Although this area participates in single-word reading [64], its presence in this contrast indicates decision making as it was reported in [22, 23].

Finally, the study of the inverse contrast c) congruent > incongruent did not show any significant activation. This indicates that the major changes in the brain activation pattern are introduced by the incongruent stimuli.

Therefore, taking into consideration all the significant activated areas, we conclude that incongruent combinations of word+emoji emotionally charged led to longer reaction times in memory-retrieval process and to activate networks related to attention and conflict resolution. These results agree with similar fMRI studies in the Stroop effect [22–24], where the authors investigated the different neuronal responses of the brain to incongruent word-color stimuli.

### Face recognition

Previous studies in fMRI with emoticons [18, 19] did not find significant evidence to support the activation of face-emotion recognition networks. However, their role within the AMT and words processing are still unknown.

In this study, we also investigated the role of emojis regarding face-emotion recognition within the memory-retrieval process. The results from the contrast word+emoji > pseudoword revealed significant activation within the inferior frontal gyrus, amygdala, and right temporal pole. Several studies have shown the role of the inferior fusiform gyrus (BA37) to familiar-face recognition [72, 73], whereas the amygdala also rapidly reacts to salient stimuli, such as faces

with emotional expressions [74–76]. Furthermore, the temporal pole plays a key role in the cognition of emotions from emotional images and faces [65, 67].

Although these observations seem to suggest the activation of the face-recognition network, the investigated contrasts do not allow us to conclusively confirm this hypothesis, since the implemented task presents words and emojis simultaneously. Further investigations, using alternative tasks, for example, including only words or emojis separately, could help to isolate the effects of emojis from words and shed light on this question.

### BTD results

The BTD analysis shows that the activation patterns correlated with the word+emoji experimental task overlap with areas from the contrast word+emoji > pseudoword of FSL (see Fig 4). In particular, BTD shows significant activation within the ACG, Broca's area, inferior frontal gyrus (BA 47) and left temporal pole, which support our findings regarding FSL's analysis.

Unlike the FSL's results, BTD showed significant activation within the posterior cingulate gyrus, which is known to participate in AMT [21, 77, 78], emotional semantic process [79] and face recognition [80]. This finding contrasts with the results reported in [19], where no significant activation was detected within the posterior cingulate gyrus, concluding that emoticons may not have a clear semantic meaning. Besides, BTD results do not show significant activation within the inferior fusiform gyrus (BA37).

On the other hand, the areas correlated with the pseudoword task show significant activation within the superior parietal lobe. In particular, we have found significant activation within the BA7 and BA40, which corresponds to brain regions that participate in single-letter reading, recognition and integer calculation [81, 82].

Interestingly, we did not observe significant activation within the posterior cingulate for the areas correlated with pseudowords, whereas we did for the areas correlated with the word +emoji combinations. This result contrasts with the observed results of FSL, where the contrast word+emoji > pseudoword did not exhibit significant activation in this area. However, a closer examination of this contrast reveals significant activation in this area after reducing the threshold to $z > 2.32$ ($p < 0.01$, uncorrected). After this observation, we presumed that the low number of participants in this study may have obscured the presence of this area within the FSL's contrast, and evidences the robustness of BTD on the aforementioned region even with a smaller number of subjects.

### Limitations of our study

The main limitation of our study is the relatively small number of subjects. Furthermore, the use of a pseudo-block design (that facilitates the use of BSS methods), though leading to higher detection power, can be proved risky because low frequency signals such as the scanner drift can become correlated with the task conditions. Multiple transitions between the same conditions and rules for the randomness of the presentation of the stimuli have been included to account for such difficulties but still the problem may exist. A future study, with a larger number of participants, focusing on the emotional content of the memory retrieved, may complement the results obtained in this study. A detailed psychometric interview (similar to the one used in [35]) of a larger sample of subjects would involve alternative regressors based on the type of the memory retrieved.

### Conclusions

In this paper, the influence of the emojis in combination with words on the autobiographical memory retrieval and the effect of incongruent emotionally charged combinations was

studied. To the best of our knowledge, this is the first time that the impact of emojis on the autobiographical memory retrieval is documented in fMRI.

The study of fMRI data have shown that, in contrast to pseudowords, word+emoji combinations activated the left Broca's area (BA44 and BA45), the amygdala, the right pole (BA48), and several frontal regions such as the SMA and the IPC, which are associated with several functional brain networks such as word-processing, emotion recognition and memory retrieval. Although the results of this study may also provide evidence for the implication of face-emotion recognition areas during the memory retrieval induced by the emojis, we cannot confirm the activation of face areas even though FSL's contrasts showed significant activation within the BA37 and BA38.

On the other hand, the study of the effect of incongruent vs. congruent word+emoji combinations revealed that emojis convey an emotional content strong enough to increase the reaction times and the number of omissions from the participants, while also activating brain regions related to the attention network and language processing, such as ACG, the inferior frontal gyurs (BA47) and Broca's area.

fMRI data analysis using BTD, which has only recently been proposed for task-related studies, has revealed similar areas with the results obtained from FSL. The study of the activated areas correlated with word+emoji combinations has confirmed the presence of language, emotion and memory-retrieval processing. Interestingly, unlike FSL's contrast, BTD shows significant activation within the posterior cingulate gyrus, an area that participates in words and face recognition process. On the other hand, the BTD map did not show significant activation within the inferior fusiform gyrus (BA37). Finally, the study of the areas correlated with the pseudoword task showed significant activation within BA7 and BA40, which corresponds to expected brain regions that participate in single-letter reading and recognition and integer calculation [81, 82].

A limitation of our study is the relatively small number of subjects. A repetition of the experiment, with a larger number of participants, may complement the reported results. The full dataset is available at https://openneuro.org/datasets/ds002711.

## Supporting information

**S1 Text. Supplementary material.** Extra file with supplementary information regarding the presented study.
(PDF)

**S2 Text. Consent form.** The consent form which was signed from the volunteers in the study.
(PDF)

**S1 Fig.**
(TIFF)

## Acknowledgments

The authors would like to thank Dr. M. Papadatou-Pastou, National and Kapodistrian University of Athens, Greece, for providing the results of the statistical analysis for the selection of the words used in the paper, Prof. A. Protopapas, University of Oslo, Norway, for his critical comments on this paper, the personnel of the imaging department of Bioiatriki S.A. at Ampelokipoi, Greece, and all the volunteers who have participated in this study. Last but not least, the authors would like to thank the anonymous reviewers and the editor, whose comments helped improve the quality of the final manuscript.

## Author Contributions

**Conceptualization:** Christos Chatzichristos, Nikolaos Andreadis, Yiannis Kopsinis.

**Data curation:** Manuel Morante.

**Formal analysis:** Christos Chatzichristos, Manuel Morante.

**Investigation:** Christos Chatzichristos.

**Methodology:** Christos Chatzichristos, Nikolaos Andreadis.

**Resources:** Nikolaos Andreadis.

**Supervision:** Eleftherios Kofidis, Yiannis Kopsinis, Sergios Theodoridis.

**Visualization:** Christos Chatzichristos.

**Writing – original draft:** Christos Chatzichristos, Manuel Morante.

**Writing – review & editing:** Eleftherios Kofidis, Sergios Theodoridis.

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
