## [Decision Letter · Decision Letter 0]

15 Jan 2020

PONE-D-19-20481

Emojis influence autobiographical memory retrieval from reading words: An fMRI-based study

PLOS ONE

Dear Mr Chatzichristos,

Thank you for submitting your manuscript to PLOS ONE. After careful consideration, we feel that it has merit but does not fully meet PLOS ONE’s publication criteria as it currently stands. Therefore, we invite you to submit a revised version of the manuscript that addresses the points raised during the review process.

I apologize for the extended time it took to reach a decision on this manuscript. We had numerous declines for review, and also numerous cases where a reviewer agreed and never submitted the review despite numerous requests. As I said in previous correspondence, the typical rules at PLOS One require two reviewers, however, because you've waited so long for a decision, I've decided to personally serve in a review capacity here to help you get this into shape for PLOS. Should you answer my and the other reviewer's comments in a satisfactory manner, I hope to not need to include an additional reviewer. Part of the trouble I believe with finding reviewers for this is that many of the people studying emojis don't have the requisite fMRI experience (or vice versa). Further, there are large gaps in the literature review and the alignment of the results with theory in this area, as pointed out by myself and the reviewer, that probably led to some of the reviewer attrition we experienced. Please thoroughly address all of my points (my review is below in additional editor comments) and the anonymous reviewer's concerns before resubmitting the manuscript. 

We would appreciate receiving your revised manuscript by Feb 29 2020 11:59PM. To enhance the reproducibility of your results, we recommend that if applicable you deposit your laboratory protocols in protocols.io, where a protocol can be assigned its own identifier (DOI) such that it can be cited independently in the future. For instructions see: http://journals.plos.org/plosone/s/submission-guidelines#loc-laboratory-protocols

We look forward to receiving your revised manuscript.

Kind regards,

Tyler Davis, Ph.D.

Academic Editor

PLOS ONE

Additional Editor Comments:

In addition to all of the points made by the first reviewer, I would like you to address the following in your revision:

Abstract- "and trigger functional brain networks related to face recognition, contrary to emoticons"

The study doesn't look at emoticons, so this isn't a conclusion that can be reached here.

Introduction - the introduction contains a considerable amount of material that is superfluous to any of the brain or behavior predictions. Some info on the history of emoticons is (perhaps) useful for setting the stage for the rest of the paper, but the first 3 paragraphs include a lot of information about emojis and emoticons that aren't necessary. This comes at the seeming expense of discussing the relatively wide research on emotional congruence on fMRI activation and memory performance, or why such a question would be asked in the first place. Likewise there is a considerably amount of work done to describe differences between fMRI and EEG that are more encyclopedic (spatial vs temporal resolution trade-offs) than they are justifying the study per se. Finally, there is a decent amount of hopping back and forth between EEG and fMRI work on emoticons or emojis, but how exactly it motivates the present work is not always clear. I would like to see the introduction substantially reworked to connect better with the affective science literature on emotional congruence and effects of emotion and emotional congruence on memory. It would also be good to more tightly integrate the previous emoticon and emoji research with the affective science research on emotional congruence when describing the motivations for the present study. Simply put, the study needs to be more clearly integrated with the existing literature and better motivated.

Methods- missing some parameters for your EPI acquisition, such as flip angle

" we defined three different contrasts: incongruent

(HS+SH) vs. congruent (HH+SS), congruent vs. incongruent and word+emoji vs. pseudowords (pseudowords have been used as the baseline regressors)"

Ive read this a bunch of times and I am having a hard time figuring out what was actually tested. The contrasts should be clearly enumerated. Is it 1) congruent vs incongruent 2) congruent and incongruent vs pseudo words 3) word+emoji vs pseudo words?

"Although the initial assumption was that the congruent vs. incongruent contrast could have also provided valuable information, no significant activated voxels were detected at the single-subject level.

Therefore, based on the findings of the first-level analysis and the assumptions made, only two contrasts were taken into consideration at the second (group) level:"

This isn't at all proper procedure in fMRI analysis (testing whether an effect exists at level one before moving to a second level). For one, it is a biased analysis as it uses information about the contrast to select results to report (see Kriegesorte's paper on double dipping). Second, the within subject variance is the only variance of interest in the first-level model, but plays a minor role in the second level model where the between subject variance is the primary variance of interest. Thus the only test that makes any sense to look at is the second level analysis. If a contrast is of interest at any level, it needs to be examined and reported at the second level.

No criteria for significance is listed in the imaging methods, so it is also difficult to assess whether anything was or wasn't conventionally significant within subjects

There is a lot of work being done in the methods to justify the blind source separation. To the degree that this is novel and a major part of the present work, it needs to be covered prior to the methods, in the intro. In the methods, the implementation and software, etc. should be described but a long description of the method and justification of it relative to the GLM is less appropriate. From my perspective, there isn't much new happening here as tensor ICA is implemented in many commonly available fMRI analysis packages. Perhaps the BTD algorithm is less frequently used. In either case, this section could be streamlined to convey what was done as opposed to having extensive explanation of how the method works and justification for it. To the degree those are necessary, they should be moved to a more appropriate section.

-Results The ANOVA, the t-test, and Tukey test are somewhat redundant. It's conventional to report an omnibus ANOVA prior to a post hoc test for a difference between means (Tukey or t-test). Here it seems to make the most sense to simply report the Tukey test after the ANOVA as it is clearer about what is driving the observed ANOVA and difference between congruent and incongruent than the first reported t-test.

The fMRI analysis starts by saying that it is reporting tfce corrected results at a threshold of p < .05 but then the figure reports z > 3, which is reported as p < .01, corrected. I don't know how to reconcile those two things but z > 3 is closer to a one-tailed uncorrected p of p < .001 (~3.09). If standard tfce p < .05 is used, then that should be what is depicted in the figure. Likewise, the table reports significant clusters with k > 20, but that again, isn't really how tfce works, which gives voxel-wise corrected p-values. Any cluster that meets the cut-off chosen in the study should be reported.

I'm confused again about what was actually tested with respect to Fig 4 as the methods state that the congruent vs non-congruent contrast was abandoned due to not finding first level results. However, Fig 4 states that panel a is congruent vs non-congruent.

For the GICA and BTD results, it's not clear why tfce wasn't used and the z > 3 (p < .01) issue arises again. Typically one would threshold all whole brain maps in a study using approximately the same criteria. Z > 3 and 20 voxels alone isn't a valid correction for multiple comparisons; something else needs to be used.

Discussion-

Need clearer limitations - 15 subjects is the total sample size and is small given current standards in fMRI research. Using the same sequence for all or most subjects, particularly one with a slow block design like the current is risky because it makes consistent low frequency information like scanner noise correlate with the signal. Thus although using the same sequences is necessary for tensor decomposition, it needs to also be listed as a possible limitation. It's also not clear how the contrasts tested and the design of the task lend themselves to studying emojis per se as emoticons or other pictorial emotional stimuli are not used. Emojis are only compared to words and pseudo words, which means that many of the differences may reflect simple differences in language-based vs pictorial representations and congruent vs incongruent emotional stimuli as opposed to anything specific to the representation of emojis.

"emojis manage to trigger also particular face-emotion recognition areas."

As per the abstract comment, this can't be stated with any certainty here as it revolves around what Poldrack refers to as a "reverse inference". The fusiform represents a lot of different things, and without an independent localizer showing that this is the fusiform face area, it's impossible to say with any certainty that the emojis are triggering a face specific region.

Despite all of the effort to explain and justify use of the ICA approaches, it's not clear what conclusions one should reach from them in the discussion or why they were done (other than to do something different from standard fMRI analysis). As per above, the theoretical merit (if any of these analyses) needs to be clearer and connect to the conclusions/discussion (as well as the intro). With the small discussion as is, I don't really understand what these methods add in this case as they aren't substantially different for any of the contrasts where emojis matter.

The discussion should again touch on how these results relate to emotional congruence literature more broadly.

Minor:

There are a number of language issues. To remedy this, I would suggest recruiting an english language speaker for proofreading.

Some sentences don't make sense to me and need to be more justified. For example:

"attempt to release the imagination of the subjects."

What evidence is there that providing word combinations would do this? If none, why is it mentioned? Later it becomes clear that the manipulation asks people to recall a memory associated with the two. In that case, it seems like it's not so much to release the imagination of the subjects and instead part of the design as people are more likely to be able to construct a memory from two words than from a full sentence.

" till"

Is used instead of until in a few places

"who missed to respond"

Failed to respond to, or missed responding to would be more appropriate wordings

" can help into establishing "

Can help to establish

"on the process of the memory retrieval"

On the process of memory retrieval

"Correlated to"

Should be 'correlated with' as correlation is non-directional

"lower amplitude"

Less activation?

"Eventually,"

Word choice. This makes it sound like you had to do a bunch of different analyses to find the activation, whereas it simply is the last area you are discussing from a single analysis

'The research leading to these results was funded by the European Union's H2020 Framework Programme (H2020-MSCA-ITN-2014) under grant agreement No.~642685 MacSeNet. The funders had no role in study design, data collection and analysis, decision to publish, or preparation of the manuscript.'

We note that one or more of the authors are employed by a commercial company:

Bioiatriki SA and Libra MLI Ltd

Reviewers' comments:

Reviewer's Responses to Questions

**Comments to the Author**

1. Is the manuscript technically sound, and do the data support the conclusions?

Reviewer #1: Yes

2. Has the statistical analysis been performed appropriately and rigorously? 

Reviewer #1: Yes

3. Have the authors made all data underlying the findings in their manuscript fully available?

Reviewer #1: Yes

4. Is the manuscript presented in an intelligible fashion and written in standard English?

Reviewer #1: Yes

5. Review Comments to the Author

Reviewer #1: The focus of this paper is timely, important, and likely to be of keen interest to a broad spectrum of psychology fields. In the attachment, I provide a few suggestions for further strengthening the manuscript.

6. PLOS authors have the option to publish the peer review history of their article (what does this mean?). If published, this will include your full peer review and any attached files.

Reviewer #1: No

---

## [Author Response · Author response to Decision Letter 0]

9 Mar 2020

The answers to the reviewers have been attached in a file.

---

## [Editor Report · Decision Letter 1]

31 Mar 2020

PONE-D-19-20481R1

Emojis influence autobiographical memory retrieval from reading words: An fMRI-based study

PLOS ONE

Dear Mr Chatzichristos,

Thank you for submitting your manuscript to PLOS ONE. After careful consideration, we feel that it has merit but does not fully meet PLOS ONE’s publication criteria as it currently stands. Therefore, we invite you to submit a revised version of the manuscript that addresses the points raised during the review process.

There are a number of minor clarification and language issues that should be corrected before acceptance of the manuscript. I'd encourage employing the services of an English language expert or copy editor as well before re-submitting as there are a number of sentences throughout the manuscript that are difficult to parse and could benefit from some fine tuning. The description of stroop tasks in the introduction is particularly unclear, for example.

Specific comments:

"and trigger several functional brain networks related to word processing"

Trigger isn't an appropriate description here. Perhaps just say that emojis lead to activation in the specific brain regions that are found to be activated in the study. As discussed in my previous review, in fMRI reporting we like to keep the amount of "reverse" inferences" to a minimum, perhaps outside of discussion. Here many readers would have questions about what exact brain regions are activated for word processing, emotion cognition, and memory retrieval as many regions do these things. As discussed before, it also isn't possible to say that these regions were doing these things in this context because none of the contrasts really specifically isolate these functions. Thus, particularly in the abstract, the actual results need to be spelled out. Not only should the specific brain regions be written here, but the specific test should be listed before describing the conclusions, like "Compared to pseudo words, words and emojis activated regions x, y, z" "There was more activation in regions x, y, z for congruent compared to incongruent emoji and word combinations" These will help readers understand what was actually found and make correct inferences about the results.

"delays in the memory retrieval"

No "the" is needed. Just delays in memory retrieval. However, it would be more accurate to say that it led to longer reaction times. Like above, the conclusion that this is memory retrieval delays per se is not a sound inference.

The authors should replace the words neurological or neurology with neurophysiological or neurobiology in the majority of places in the manuscript. neurology is a specific medical discipline, which focuses brain pathology, and thus saying "neurological" gives the impression that patient work might be presented, which it is not. The current manuscript is more in line with neurophysiology, neuroscience, neurobiology, or neuroimaging (any of which would be more appropriate to list than neurological)

neurological findings -> neurophysiological data

neurological -> neurophysiological (in many places throughout the manuscript)

"Several EEG studies have pointed out the emotional content of emoticons and emojis"

It seems like emotional content of emoticons and emojis is somewhat of a given that doesn't need imaging to establish. Is the point of this sentence that" EEG studies have identified neurobiological processes associated with processing emotion in emoticons and emojis"?

"whether the emoticons are processed"

whether emoticons are processed

"iii) since no significant activity

was detected at the fusiform gyrus or the posterior cingulate gyrus,emoticons do not

carry clear semantic content, in contrast to nouns or adjectives."

This isn't really a valid inference as fusiform and posterior cingulate gyrus aren't necessarily the only semantic processing regions (or even the most likely semantic processing regions) and lack of activation differences are like any other statistical test; it doesn't mean you can accept the null when a brain region doesn't activate

"The initial assumption of our

study is that since the emoticons influence the reading of complete sentences, they will

also influence the memory retrieval process since it is documented that they have also

emotional content."

Clause structure of this sentence is complex - there are two "since"s in it. Should be simplified or split into two.

In the fmri acquisition section, there appears to be a paragraph more about the fmri task procedure. That should go in the section with the task.

There is still inconsistency in the z's listed as significant, even within the FEAT analysis section. For example, one says z > 3.09 the other says z > 3.01, the former is closer to the standard p < .001 one-tailed cutoff for the z-distribution. The authors should use a single z cutoff.

"Nor a closer examination over the different subjects at the single-subject level did provide significant results for any of the participants."

Meaning of sentence unclear

"associated to"

associated with

"tasks of interests"

interest without an s

For fmri results, activation is the preferred term instead of "activity"

For example:

"First of all, the maximum significant

activity of this contrast appears within the Broca's area (left BA44 and BA45),"

The peak of maximal activation for the contrast of words to pseudo words was observed in Broca's area

"The BTD analysis shows that the activation patterns correlated with the word+emoji experimental task are similar to"

similar to might be better described as "overlaps with" as no direct test of similarity was presented

"correlated to"

correlated with

"correspond to"

corresponds to

"z > 2.32"

Why was z of 2.32 chosen? This doesn't correspond to any of the defaults in FSL where a z of 2.3 is conventionally used as the default threshold

"been documented in fMRI"

documented with fMRI

"trigger several functional brain networks"

Are associated with greater activation in several functional brain regions.

We would appreciate receiving your revised manuscript by May 15 2020 11:59PM. To enhance the reproducibility of your results, we recommend that if applicable you deposit your laboratory protocols in protocols.io, where a protocol can be assigned its own identifier (DOI) such that it can be cited independently in the future. For instructions see: http://journals.plos.org/plosone/s/submission-guidelines#loc-laboratory-protocols

We look forward to receiving your revised manuscript.

Kind regards,

Tyler Davis, Ph.D.

Academic Editor

PLOS ONE

---

## [Author Response · Author response to Decision Letter 1]

3 May 2020

The response to the reviewers is attached in a separate pdf file

---

## [Editor Report · Decision Letter 2]

20 May 2020

Emojis influence autobiographical memory retrieval from reading words: An fMRI-based study

PONE-D-19-20481R2

Dear Dr. Chatzichristos,

We are pleased to inform you that your manuscript has been judged scientifically suitable for publication and will be formally accepted for publication once it complies with all outstanding technical requirements.

With kind regards,

Tyler Davis, Ph.D.

Academic Editor

PLOS ONE
---

## [Editor Report · Acceptance letter]

5 Jun 2020

PONE-D-19-20481R2 

Emojis influence autobiographical memory retrieval from reading words: An fMRI-based study 

Dear Dr. Chatzichristos:

I'm pleased to inform you that your manuscript has been deemed suitable for publication in PLOS ONE. Congratulations! Your manuscript is now with our production department. 

Kind regards, 

on behalf of

Dr. Tyler Davis 

Academic Editor

PLOS ONE